# Prevalence and Antimicrobial Resistance of *Campylobacter jejuni* and *Campylobacter coli* from Laying Hens Housed in Different Rearing Systems

**DOI:** 10.3390/ani12212978

**Published:** 2022-10-29

**Authors:** Gaia Casalino, Giancarlo Bozzo, Francesca Rita Dinardo, Francesco D’Amico, Michela Maria Dimuccio, Antonio Camarda, Edmondo Ceci, Diana Romito, Elena Circella

**Affiliations:** Department of Veterinary Medicine, University of Bari “Aldo Moro”, s.p. Casamassima km 3, 70010 Valenzano, Italy

**Keywords:** *Campylobacter jejuni*, Campylobacter coli, laying hen, rearing system, antibiotic resistance, welfare indicators

## Abstract

**Simple Summary:**

This survey investigated the incidence of *Campylobacter (C) jejuni* and *C. coli* infection in laying hens housed in farms which use different rearing systems: cages (A), barns (B) and aviaries (C). Two flocks (1 and 2) for each farm were evaluated. The hen plasma levels of corticosterone and interleukin 6 (IL-6), which are considered welfare indicators, and the sensitivity of detected *Campylobacter* strains to the antibiotics were investigated. The highest (*p* < 0.05) levels of IL-6 and corticosterone were found in laying hens housed in aviaries. *C. jejuni* and *C. coli* were identified in 77/177 (43.5%) and 69/177 (38.9%) laying hens, respectively. *C. jejuni* and *C. coli* were simultaneously detected in 14 laying hens (7.9%). *C. jejuni* was prevalently found in laying hens housed in barns (B1: 53.3%; B2: 46.7%) and aviaries (C1: 34.6%; C2: 86.7%). *C. coli* was prevalently identified in laying hens housed in cages (A1: 41.9%; *p* < 0.001; OR: 10.11, CI_95%_: 2.04–50.19; A2: 80%; *p* < 0.001; OR: 56.00, CI_95%_: 10.33–303.68) and, depending on the flock, in barns (B2: 40%) and aviaries (C1 54.8%). In particular, antibiotic resistance to fluoroquinolones and tetracycline was found, and it was mainly detected among *C. coli* strains. Multidrug resistance was found in 19.7% of *C. coli* and 17.5% *C. jejuni* strains, respectively. The incidence of *Campylobacter* infection found in the farms highlights the opportunity to increase the biosecurity measures to be adopted in the management of laying hen flocks.

**Abstract:**

*Campylobacter (C.) jejuni* and *C. coli* are responsible for food poisoning in humans. Laying hens may host the bacteria usually without developing symptoms. The aims of this paper were to evaluate the incidence of *C. jejuni* and *C. coli* infection in laying hen flocks housed in different rearing systems, the plasma levels of two welfare indicators (corticosterone and interleukin 6, IL-6) and the antimicrobial resistance of the detected *Campylobacter* strains. Two different flocks (1 and 2) from cage (A), barn (B) and aviary (C) farms were investigated. The highest (*p* < 0.05) levels of IL-6 were detected in laying hens housed in aviaries. A similar trend emerged in corticosterone level, although differences were found between C1 and C2. *C. jejuni* and *C. coli* were identified in 43.5% and 38.9% of birds, respectively. In total, 14 out of 177 (7.9%) hens simultaneously hosted *C. jejuni* and *C. coli.*
*C. jejuni* was prevalently detected in hens housed in barns (B1: 53.3%; B2: 46.7%) and aviaries (C1: 34.6%; C2: 86.7%). Conversely, laying hens housed in cages were significantly exposed to infection of *C. coli* (A1: 41.9%; A2: 80%) while, regarding barns and aviaries, a significant prevalence emerged only in flocks B2 (40%) and C1 (54.8%). Simultaneous infection was statistically significant in barn B1 (36.7%). Antibiotic resistance was mainly detected among *C. coli* strains, and it was most frequent for fluoroquinolones and tetracycline. Multidrug resistance was also found in *C. jejuni* (19.7%) and *C. coli* (17.5%) strains. Based on the results of this study, we recommend increasing biosecurity and hygienic measures to manage hen flocks.

## 1. Introduction

In recent times, the consumption of safe foods and their impact on human health have changed the concept of food safety and radically reshaped it. Consumers’ attention is focused on types of farming that ensure the hygiene of animal products, but at the same time, concerns about the ethical aspects of animal-sourced foods have increased. Animal welfare and human health continually fuel interest in animal-friendly farming systems [1,2]. As a result, the conventional cage system is being replaced by cage-free and free-range production systems to ensure animal healthiness and improve nutritional features of products [3]. Currently, Legislative Decree 27 September 2010 n. 181 establishes minimum standards for the protection of chickens reared for meat production in Italy. Legislative Decree 29 July 2003 n. 267 establishes the protection of laying hens and the registration of their breeding systems, by allowing the farming of birds in enriched cages and cage-free systems, in aviaries or on litters. However, beginning in 2023, the phasing out of cages by 2027 was planned to encourage a complete transition to litter-breeding systems. In addition to ethical reasons, reducing stress in hens will also have health implications [4,5]. An ideal balance between egg production and laying hens’ welfare is the free-range system [6], but this technique is potentially linked to microbiological implications that remain controversial [7]. De Reu et al. [8] found no systematic differences in Gram-negative bacteria distribution on eggshells from laying hens reared in conventional cages, furnished cages and aviary systems. Later, the same authors reported that bacteriological egg contamination and eggshell quality differed substantially among individual farms raised under the same housing system [9]. These results seem to indicate that other factors such as flock management, shed structure or hygienic conditions may influence bacterial eggshell contamination in addition to the breeding system [9]. Biosecurity measures and the hygienic level of farms are critical points in the poultry industry as they can improve the overall flock health, prevent disease, decrease the cost of treatments, reduce losses, and improve farm profitability [10]. Although *Clostridium perfringens*, *Salmonella enterica* subsp. *enterica*, *Listeria* spp., *Escherichia coli*, *Campylobacter* spp. and *Vibrio vulnificus* are the major bacteria leading to foodborne illnesses, *Campylobacter* spp., *Salmonella* spp. and *E. coli* are the ones closely associated with poultry [11,12]. Hens’ resistance to infections by *Campylobacter* spp. and *Staphylococcus* spp. seems to be inversely linked to their stress levels [4,5].

Bacteria belonging to the *Campylobacter* genus are commensals of gut microflora in animals. They are small (0.2–0.8 μm × 0.5–5 μm), Gram-negative and slender spirally curved rods. With the exception of *C. gracilis, Campylobacter* spp. are equipped with single polar unsheathed flagellum at one or both ends of the cell or multiple flagella (*C. showae*) that confer a corkscrew-like movement. *Campylobacter* species are thermotolerant and grow in a temperature range of 37 to 42 °C, with an optimum growth temperature at 41.5 °C. This thermophilic nature may explain the high prevalence rate found in poultry. They are a microaerophilic organism and grow better at low oxygen tension (5% O_2_, 10% CO_2_ and 85% N_2_) [13].

Campylobacteriosis is the most common food poisoning in Europe and worldwide, and it has been prevalently linked to poultry meat consumption. Nevertheless, more than 50% of cases in humans may be attributed to the handling of and contact with chickens which are a reservoir for *Campylobacter* [14]. *Campylobacter*
*jejuni* and *C. coli* are the most identified species in fecal samples from human patients with gastroenteritis due to *Campylobacter*. According to FoodNet, 89% of campylobacteriosis cases in humans are induced by *C. jejuni*, 8% by *C. coli*, and 3% by other species. *Campylobacter* in humans causes gastrointestinal disease, usually self-limiting, lasting 5 to 7 days. Symptoms typically occur 2 to 5 days after ingestion of the bacteria and include diarrhea with cramping, acute abdominal pain, nausea and vomiting. In infants and young children, bloody diarrhea without fever may be the only clinical manifestation [15]. In many cases, the infection resolves without antimicrobial treatment. Instead, hospitalized patients are treated with fluoroquinolones and macrolides, which are the drugs of choice for treating *Campylobacter* infections [16]. In 5–10% of patients, especially if immunocompromised, elderly or pregnant women, extra-intestinal infections may develop, including Guillain–Barre syndrome (GBS), Miller Fisher syndrome and reactive arthritis (ReA). GBS, due to *C. jejuni*, is an autoimmune disease of the peripheral nervous system (PNS), characterized by acute paralysis that can lead to respiratory muscle impairment and consequently cause patient death [17]. Miller Fischer syndrome is considered a rare variant of GBS [18], characterized by the triad of ataxia, areflexia and ophthalmoplegia [19]. Reactive arthritis, associated with *C. jejuni* infection, is usually an asymmetric oligoarticular arthritis involving the knees, ankles or wrists and occurring about 10 days after clinical manifestation of enteritis [15].

*Campylobacter jejuni* and *C. coli*, which are commonly considered commensal germs of poultry gut, especially in the presence of predisposing conditions such as lack of hygiene in flock management, overcrowding and hierarchic struggles, are known to invade the tissues, particularly the liver, causing avian vibrionic hepatitis (AVH) [20]. AVH is characterized by 1–2 mm multifocal, grayish-white or cream-colored liver lesions, increased mortality in the flock (up to 10%) and reduced egg production (10–25%).

Recently, a new *Campylobacter* species, *C. hepaticus*, has been identified as responsible for Spotty Liver Disease (SLD) in laying hens [21,22]. SLD is an emerging disease in Europe, Australia and the United States, particularly in outdoor and free-range farms, and is characterized by lesions and decreased egg production very similar to those observed in AVH.

Colonization of poultry intestine by *Campylobacter* occurs during the rearing period and it is age-dependent. The microbiota changes during the growing of the animal and has a deep impact on the gut immune system [23]. Broiler chickens less than two weeks old are rarely colonized, and *Campylobacter* is more frequently detected in the last period of growing before slaughter [24].

The higher incidence of *Campylobacter* infection in laying hen farms compared to broiler flocks depends on the age of the hens and their longer production cycle, potentially exposing them to numerous sources of contamination [25,26,27,28]. The main mode of transmission of *Campylobacter* among birds is the fecal–oral route, so hens reared in cage-free systems are potentially more susceptible to infection. As a result, the incidence of SLD seems to be lower in hens reared in cages than in cage-free systems [21]. In positive farms, the isolation of *Campylobacter* spp. or the detection of DNA of *Campylobacter* can be made in environmental samples, including water, soil, dust, mites and rat feces. Moreover flies, which may be involved in *Campylobacter* transmission within flocks, have been recently identified as possible vectors of *C. hepaticus* [21].

Corticosterone secretion from the adrenal gland cortex of birds is stimulated by adrenocorticotropic hormone (ACTH) from the pituitary gland, which in turn is stimulated by corticotropin-releasing factor (CRF) and arginine vasotocin (AVT) from the hypothalamus [29]. The hypothalamic–pituitary–adrenal (HPA) axis is activated in response to stressors, with an increase in plasma corticosterone concentrations [30]. Corticosterone levels have been previously assessed as a stress response in studies of fear behaviors in chickens and Japanese quail (*Coturnix coturnix japonica)* [31]. In plasma, the simultaneous increase in corticosterone [32] and pleiotropic cytokine IL-6 [33] may be suggested as an adaptive physiological body response involving the HPA axis and the immune system to stressors [34]. Considering the recent changes in laying hens’ rearing systems, the aims of this study were to evaluate: (i) the incidence of *C. jejuni* and *C. coli* infection in laying hens reared in different housing systems; (ii) the antimicrobial resistance of the detected *Campylobacter* strains; and (iii) the plasma levels of corticosterone and interleukin (IL-6).

## 2. Material and Methods

### 2.1. Study Design and Sampling

The study was carried out in three laying hen farms using enriched cages (A), barns (B) and aviaries (C), respectively. All farms were located in southern Italy, in the provinces of Taranto (A) and Lecce (B and C). Two different flocks (A1, A2, B1, B2, C1, C2) were tested on each farm. All animals were fed with commercial feed. Each shed was equipped with automatic control systems to provide constant environmental temperatures. In total, 177 laying hens were tested. Details on the bird flocks and the number of individuals tested are reported in Table 1.

A cloacal swab was collected from each bird. The blood sample from ulnar vein was also collected to evaluate two animal-based measures (ABMs): plasma corticosterone and interleukin (IL-6). Bird-handling and sampling were performed according to the guidelines of the Ethics Committee for Animal Experimentation of the Department of Veterinary Medicine (DiMeV), Bari, Italy (Approval number 21/2021).

All collected samples were transferred within two hours after sampling, under refrigerated conditions to the Department of Veterinary Medicine of Bari, Southern Italy.

### 2.2. Campylobacter Identification

Cloacal swabs were placed in sterile tubes containing 5 mL of Nutrient Broth (Oxoid, Basingstoke, UK) supplemented with 5% sheep blood (Oxoid), Campylobacter Selective Supplement SR0085E (Oxoid), and Growth Supplement SR0232 (Oxoid), and were incubated at 42 °C under microaerobic conditions for 24 h. The samples were plated on Campylobacter agar base (Oxoid), supplemented with 5% sheep blood (Oxoid), Campylobacter Selective Supplement SR0098E (Oxoid), and Campylobacter Growth Supplement SR0232 (Oxoid). The plates were incubated under the above described same conditions for 48–72 h. Colonies morphologically compatible with *Campylobacter* spp. were transferred onto blood agar plates (Oxoid), and incubated at 42 °C under microaerobic conditions for 24 h. Of each suspected isolated, three colonies were tested by multiplex PCR according to Denis et al. [35], with modifications. Briefly, DNA extraction was performed by dissolving each colony in 100 µL of distillate water and boiled at 100 °C for 10 min.

The reaction mixture consisted of iTaq buffer 10X, MgCl2 50 mM, dNTPs 10 mM (of each of the four oligonucleotides), 11 μM of MD16S1 and MD16S2 and 10.42 μM of the remaining two primers pairs (Table 2), 1.34 U of iTaq DNA polymerase Platinum II Green HS PCR MM (Invitrogen, Lithuania) and 2 μL of sample DNA and sterile distilled water to complete a total volume of 25 μL.

Cycling conditions were as follows: 94 °C for 5 min for 1 cycle; 94 °C for 15 s, 60 °C for 15 s, 72 °C for 10 s for 34 cycles; and 72 °C for 10 min for final elongation. PCR products were loaded for electrophoresis on a 1.5% agarose gel stained with ethidium bromide. The rection was visualized with Gel Doc-It image analyzer (UVP, Upland, CA, USA). *C. jejuni* ATCC 29428 and *C. coli* ATCC 33559 obtained from LGC Promochem (LGC Promochem, Teddington, UK) were used as positive controls (Figure 1).

### 2.3. Antibiotic Susceptibility Testing

Overall, 76 strains of *C. jejuni* and 63 strains of *C. coli* were tested to determine susceptibility to azithromycin (AZM) 15 µg (Oxoid); chloramphenicol (CHL) 30 µg (Oxoid); ciprofloxacin (CIP) 5 µg (Oxoid); enrofloxacin (ENR) 5 µg (Oxoid); erythromycin (E) 15 µg (Oxoid); gentamicin (CN) 10 µg (Oxoid); acid nalidixic (NA) 30 µg (Oxoid); tetracycline (TE) 30 µg (Oxoid); and trimethoprim-sulfamethoxazole (SXT) 25 µg (Oxoid). Antibiotic susceptibility tests were performed on Mueller–Hinton agar supplemented with 5% horse blood using the standard Kirby–Bauer disk diffusion method according to the European Committee for Antimicrobial Susceptibility Testing [36,37].

### 2.4. Plasma Corticosterone and IL-6-ELISA Test

ELISA tests were performed using a DYNEX DSX^®^ fully automated four-plate ELISA processing system. Plasma corticosterone and IL-6 concentrations were measured with Avian-Corticosterone ELISA (My-Bio-Source, San Diego, CA, USA) and Avian-IL-6 ELISA (My-Bio-Source) respectively, according to manufacturer’s instructions. All reagents were kept at 25–28 °C for 30–40 min before reconstruction. Enzyme conjugate was stored at −20 °C until use. Optical density (OD) was determined using a microplate reader with a wavelength of 450 nm. The mean of the duplicate’s readings for each standard and sample was calculated, and the average OD of the blank was subtracted. A standard curve was created using computer software capable of generating a four-parameter logistic (4-PL) curve-fit.

### 2.5. Statistical Analysis

The data for different flocks were analyzed by univariate statistical analysis (Pearson’s chi-square test and Fisher’s exact test for independence) using *Campylobacter* spp. status (positive/negative) as the dependent variable. The odds ratio (OR) and 95% confidence interval (CI_95%_) were also calculated. Values of *p* < 0.05 were considered statistically significant. Statistical analysis was performed using spss 13 software for Windows (SPSS Inc., Chicago, IL, USA). Data for inteleukin-6 and cortisol levels were submitted to a one-way ANOVA to show significant differences within different flocks of birds. Data were analyzed using Statistic 13.0 (Statsoft Inc., Tulsa, OK, USA). Tukey post hoc tests was used to compare means, with significant level of 5%.

## 3. Results

### 3.1. Campylobacter Prevalence

In total, 160 out of 177 (90.4%) laying hens tested positive for *Campylobacter* (Table 3). The incidence of infection was 94.6% in laying hens reared in aviaries, 91.7% in barns and 86.7% in cage systems. *C. jejuni* and *C. coli* were identified in 77 (43.5%) and 69 (38.9%) birds, respectively. Both *C. jejuni* and *C. coli* were simultaneously detected in 14 animals (7.9%). *C. jejuni* was prevalently detected in laying hens housed in barns from both B1 (53.3%; *p* < 0.001; OR: 7.43, CI_95%_: 2.08–26.55) and B2 (46.7%; *p* < 0.001; OR: 5.69, CI_95%_: 1.59–20.33) flocks. A high prevalence was also found in aviary C2 (86.7%; *p* < 0.001; OR: 42.25, CI_95%_: 9.53–187.22), and to a lesser extent in aviary C1 (34.6%; *p* < 0.001; OR: 3.44, CI_95%_: 0.91–12.97). Conversely, laying hens housed in cages were significantly exposed to infection by *C. coli* both in A1 (41.9%; *p* < 0.001; OR: 10.11, CI_95%_: 2.04–50.19) and A2 (80%; *p* < 0.001; OR: 56.00, CI_95%_: 10.33–303.68) flocks. Regarding *C. coli* incidence in barn and aviary housing, differences were found among flocks. A significant prevalence of *C. coli* emerged only in flocks B2 (40%; *p* < 0.001; OR: 9.33, CI_95%_: 1.87–46.68) and C1 (54.8%; *p* < 0.001; OR: 16.33, CI_95%_: 3.2–83.25). Simultaneous infection was found in the cages from both flocks (A1 and A2) and in barn B1, although a statistically significant exposure was found only in the latter (36.7%; *p* < 0.001; OR: 17.37, CI_95%_: 2.07–145.61).

### 3.2. Antibiotic Resistance

Antibiotic resistance was mainly observed in *C. coli* strains (Table 4). Among *C. jejuni* strains, 14 (18.4%), 9 (11.8%) and 10 (13.1%) were resistant to ciprofloxacin, enrofloxacin and nalidixic acid, respectively, while 13 (20.6%), 10 (15.9%) and 11 (17.5%) *C. coli* strains were resistant to those drugs. In addition, partial susceptibility to these antibiotics was detected in 8 (12.7%), 16 (25.4%) and 11 (17.5%) *C. coli* strains. Twenty-five (39.7%) of the tested *C. coli* strains and four of the *C. jejuni* strains (5.3%) were resistant to tetracycline. Resistance to trimethoprim/sulfamethoxazole was detected in 10 (13.1%) and 6 (9.5%) strains of *C. jejuni* and *C. coli*, respectively.

Among *C. jejuni* strains, only one was resistant to erythromycin, while no resistance was found to azithromycin, chloramphenicol and gentamicin. No *C. coli* strains were resistant to chloramphenicol, erythromycin and gentamicin.

Drug-resistant strains of *C. jejuni* were found in the cage (A) and barn (B) systems, while no strains were detected in the aviary systems (C). Resistant *C. coli* strains were identified in all considered rearing systems but were mostly found in the aviary (flock C1 and C2).

It was found that 15 (19.7%) *C. jejuni* and 11 (17.5%) *C. coli* were resistant to more than two molecules. Details of the multidrug resistance found in the detected strains are shown in Table 5.

Resistance to ciprofloxacin/enrofloxacin/nalidixic acid association was the most frequently detected among *C. jejuni* strains and was found in strains from cage-reared birds (A:25%) (flock A1: 37.5%) but resistance to ciprofloxacin/nalidixic acid/trimethoprim-sulfamethoxazole (A:8.3%), ciprofloxacin/enrofloxacin/nalidixic acid/trimethoprim-sulfamethoxazole (B:6.9%) and ciprofloxacin/nalidixic acid/tetracycline/trimethoprim-sulfamethoxazole (B:6.9%) were also observed. Resistance to ciprofloxacin/enrofloxacin/nalidixic acid (A:15.1%) and ciprofloxacin/enrofloxacin/nalidixic acid/tetracycline (B:14.3% and C1: 16.7%) was also frequently detected in *C. coli* strains.

The cutoff values as defined by EUCAST (European Committee on Antimicrobial Susceptibility Testing) were used to interpret the antimicrobial susceptibilities to AZM: azithromycin; CIP: ciprofloxacin; E: erythromycin; TE: tetracycline. The cutoff values as defined by CLSI Clinical Laboratory and Standards Institute) were used the antimicrobial susceptibilities to CN: gentamicin; NA: nalidixic acid; SXT: trimethoprim-sulfamethoxazole CHL: chloramphenicol; ENR: enrofloxacin.

### 3.3. Corticosterone and Interleukin-6 Levels

The highest (*p* < 0.05) levels of IL-6 were detected in laying hens housed in aviaries from both C1 and C2 flocks (Figure 2). Intermediate values were found in birds raised in barns, although differences emerged between farms (*p* < 0.05). Cage housing resulted in the lowest (*p* < 0.05) levels of IL-6, showing no differences between A1 and A2.

A similar trend emerged in corticosterone level, with the highest (*p* < 0.05) values recorded in laying hens farmed in aviaries (Figure 3). Within the same housing method, significant differences (*p* < 0.05) for corticosterone amount were found only between C1 and C2.

## 4. Discussion

*Campylobacter* was detected in 90.4% of laying hens tested in the study. *Campylobacter jejuni* and *C. coli* were identified in 43.5% and 38.9% of birds, respectively. In addition, 14 hens (7.9%) were carriers of both *C. jejuni* and *C. coli.* Even if *C. jejuni* infection is responsible for avian vibrionic hepatitis [20], infected hens usually harbor the germ without developing clinical symptoms, spreading a large amount of *Campylobacter* through their feces [38,39]. Although the epidemiological dynamics of infection transmission in laying hen farms are not well known, their production cycles are longer than those used for broilers, exposing hens to close and constant contacts between infected and healthy individuals over several months [38]. This may explain why the incidence of infection can be high in laying hen farms. Infection rates of 70% [40], 90% [41], 97.9% [42] and up to 100% [43] have been reported in hen flocks from different countries, although lower incidence rates are also reported in the literature [38,43,44,45,46,47]. Several factors may influence the incidence of infection in poultry farms, and different fomites such as rodents, wild birds, visitors, equipment, drinking water and flies may carry the germ to flocks [48,49,50].

As *Campylobacter* transmission occurs via the fecal–oral route, the infection easily involves many birds [51] and could potentially spread more rapidly in cage-free flocks. Although the incidence of infection varied greatly among flocks, a prevalent trend emerged in aviaries and barns compared with cages. Similarly, a significantly higher incidence of bacterial and parasitic diseases, as well as other disorders such as cannibalism, has been previously found in laying hens reared in litter-based and free-range systems than in those kept in cages [52]. Higher levels of bacterial contamination have also been observed in eggs from cage-free systems than in eggs from caged hens [53]. In addition, more frequent antibiotics use has been reported in Germany in laying hens housed in litter-based housing systems than in caged birds [54].

The highest values of corticosterone and IL-6 were found in hens raised in cage-free systems, particularly in aviaries, while the lowest values were detected in cage-reared birds. The results seem to indicate that the rearing method may have an impact on the stress indicators evaluated. Barn and aviary systems provide animals with more movement than cages, but, on the other hand, they may increase opportunities for competition, potentially leading to increased stress and plasma corticosterone levels. The greater chances for laying hens to move around, climb the perch floors and use their wings could also explain the higher IL-6 levels found in aviaries. Indeed, IL-6 is not only a cytokine released in response to stressor situations, but also a myokine released during prolonged exercise without muscle damage [55]. Further investigations are needed to assess the actual relevance of these parameters as predictors of welfare conditions in different housing systems, and to establish a possible relationship between stress conditions and the incidence of *Campylobacter* infection in laying hens.

On the tested flocks, no relationship was found between the detection of *C. jejuni* and *C. coli* and the sampling time, except for a few flocks. Due to efficiency of ventilation systems, the temperatures recorded inside the different sheds at the time of samplings were quite similar to each other. Although 84% and 76% positivity rates have been found in Finland in autumn and in spring, respectively [43], some studies have highlighted an increased incidence of *Campylobacter* detection in chicken and pig samples at slaughterhouses in relation to increased environmental temperature [56] and increased temperature and drought on broiler farms [57]. As a result, foodborne diseases, including campylobacteriosis, occur more frequently in warmer months [58]. The higher incidence of infection in farms during summer may also be influenced by the presence of flies [59], which may carry the germ within and between flocks [60].

Regarding antimicrobial resistance, it has been found to be more frequent among *C. coli* than *C. jejuni* strains, according to some studies performed on laying hen farms [38,61]. However, in other investigations on strains from broiler [62], laying hens [47,63] and poultry meat [64], drug resistance was found more often in *C. jejuni.* Quinolones and tetracycline were the drugs to which both germs were most frequently found to be resistant. This kind of resistance was dependent on bird flock but was frequently found in *C. coli* strains from cage-free systems. Resistance to quinolones in *Campylobacter* isolated from laying hens [47,65,66], broilers [62,67,68] and chicken meat [68,69] has been found in the United States, Italy, Spain, Poland and other countries. These molecules, frequently used in veterinary medicine, are considered a viable alternative to macrolides for the treatment of campylobacteriosis in humans [66,70,71]. A relationship has been found between the use of quinolones in poultry and increased resistance in chicken and human *Campylobacter* strains [69], leading to more cautious use in the poultry industry [72]. In our study, resistance to ciprofloxacin, which is the most widely used quinolone in human medicine, was detected in *C. coli* from laying hens in barns (flock B2: 91.7%) and aviaries (flock C1: 75%), and in *C. jejuni* in laying hens from cages (flock A1: 62.5%) and barns (flock B1: 60%), according to other studies where resistance rates to quinolones other than tetracycline were relevant [61] and reached 100% [38]. Resistance to erythromycin, which is the drug of choice for the treatment of campylobacteriosis in humans, was found more rarely and was detected only among *C. jejuni* strains, in agreement with other investigations in laying hens [70,72] and broilers [68]. Resistance to tetracycline, which is often used to treat gastrointestinal disorders in humans, has been found more often in *C. coli* (39.7%) than *C. jejuni* (5.3%) and has been detected in strains from barns and aviaries. Previous studies have reported resistance rates ranging from 33.3 to 98.9% [38,47,65,67,70]. Multidrug resistance was found in both *C. jejuni* (19.7%) and *C. coli* (17.5%) and was more frequent in *C. jejuni* strains from birds reared in cages and *C. coli* isolated from barn and aviary-reared hens. Similar rates of multidrug resistance were detected in *Campylobacter* strains from poultry meat [73] and laying hens, in which 41% of multidrug resistant strains were identified [74].

Although campylobacteriosis in humans is mainly associated with the consumption and handling of chicken meat [75], the public health relevance of *C. jejuni* and *C. coli* infection on laying hen farms should be better investigated. There is no evidence of risks to humans associated with infected eggs because only eggshell contamination usually occurs, even though vertical transmission of *Campylobacter* can occur [76]. Eggshells and cuticles are effective barriers to bacteria [77,78,79,80]. Moreover, egg white contains antimicrobial proteins such as lysozyme and ovotransferrin that induce bacteriostatic effects [81]. Nevertheless, transmission of *Campylobacter* from poultry to humans can potentially occur directly by handling infected birds or indirectly during the removal of contaminated litter or bedding from cages.

Proper implementation of biosecurity measures should be able to reduce the incidence of infection in laying hen farms, as already observed in broilers [82]. In particular, the use of dedicated boots, clothing and equipment and handwashing before and after contact with the flock should be adopted [83,84,85]. The boot baths at the entrance to the shed may be effective if the disinfectant is changed regularly every week [86]. In addition, shielding ventilation openings may be useful in reducing the incidence of *Campylobacter* in flocks by preventing flies’ entry into the sheds [59].

## 5. Conclusions

Based on the results of this study, the infection of *C. jejuni* and *C. coli* occurred frequently in laying hen flocks. Resistance to quinolones, tetracycline and trimethoprim/sulfamethoxazole, as well as multidrug resistance was found in the strains tested. Although eggs do not pose a risk of *Campylobacter* transmission to humans, infection in laying hens should be considered more carefully. In fact, it usually occurs in birds without clinical signs, increasing the potential risks of transmission to humans when handling hens, removing litter or cages. Increasing biosecurity measures should be helpful in reducing the incidence of *Campylobacter* infection in laying hens’ farms.

## Figures and Tables

**Figure 1 animals-12-02978-f001:**
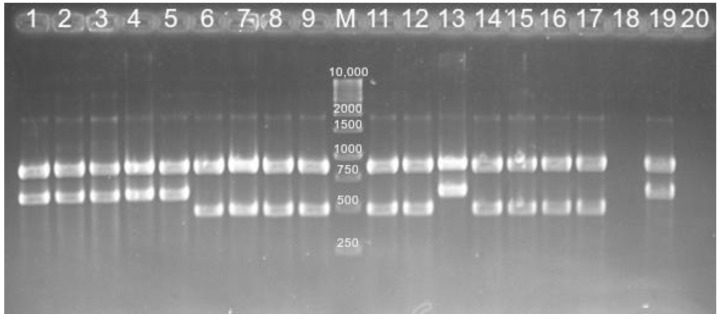
Identification of *Campylobacter* by multiplex PCR. M: Marker (O’ Gene Ruler 1kb DNA Ladder, Ready to use, 250–10,000 bp, Thermo Scientific Inc., Waltham, MA, USA); Lanes 1–5: *C. jejuni* strains; Lanes 6–9: *C. coli* strains; Lanes 11–12: *C. coli* strains; Lane 13: *C. jejuni* strain; Lanes 14–16: *C. coli* strains; Lane 17: *C. coli* positive control; Lane 19: *C. jejuni* positive control; Lanes 18 and 20: negative control.

**Figure 2 animals-12-02978-f002:**
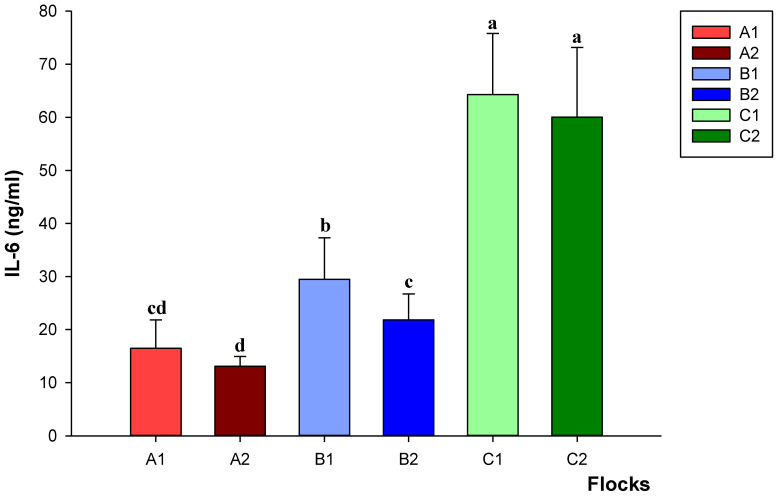
Levels of interleukin-6 (IL-6) in flocks of laying hens housed in enriched cages (A1–A2), barns (B1–B2) and aviaries (C1–C2). Data are reported as means ± standard deviations. Different letters (a–d) indicate significant differences (*p* < 0.05) among the different flocks of birds.

**Figure 3 animals-12-02978-f003:**
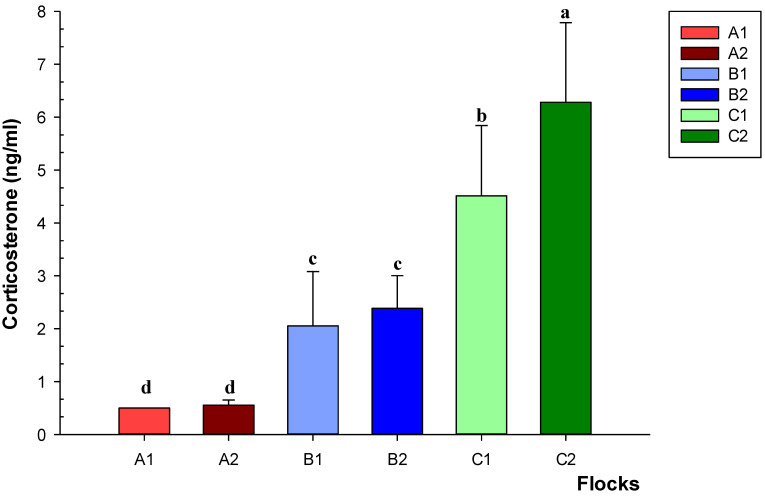
Levels of corticosterone in flocks of laying hens housed in enriched cages (A1–A2), barns (B1–B2) and aviaries (C1–C2). Data are reported as means ± standard deviations. Different letters (a–d) indicate significant differences (*p* < 0.05) among the different flocks of birds.

**Table 1 animals-12-02978-t001:** Rearing systems and details about flocks and tested individuals.

Farm	Rearing System	Flock	Consistence of Flock (Number of Birds)	Stage of Production (Months)	Environmental Temperature Inside the Shed (T °C)	Number of Sampled Birds
A	Cages	A1	12.500	6	26.9 °C	31
		A2	12.000	2	23 °C	30
B	Barns	B1	3.000	8	24 °C	30
		B2	3.000	3	20 °C	30
C	Aviaries	C1	12.000	7.5	24.7 °C	26
		C2	12.000	5	20.3 °C	30

**Table 2 animals-12-02978-t002:** Primers used for the identification of *C. jejuni* and *C. coli*.

	Target Gene	Primer	Sequence	Amplicon Molecular Weight
Genus *Campylobacter*	16S rRNA	MD16 S1 MD16 S2	ATCTAATGGCTTAACCATTAAAC GGAGGGTAACTAGTTTAGTATT	857 bp
*C. jejuni*	MapA	MD mapA1 MD mapA2	CTATTTTATTTTTGAGTGCTTGTG GCTTTATTTGCCATTTGTTTTATTA	598 bp
*C. coli*	CeuE	COL3 MDCOL2	AATTGAAAATTGCTCCAACTATG TGATTTTATTATTTGTAGCAGCG	462 bp

**Table 3 animals-12-02978-t003:** Prevalence of *C. jejuni* and *C. coli* in flocks of laying hens housed in enriched cages (A1-A2), barns (B1-B2) and aviaries (C1-C2).

	*C. jejuni*			*C. coli*			Both *C. jejuni* and *C. coli*		
Flocks	*N° Pos/Tested (%)*	*p*-Value	OR (CI _95%_)	*N° Pos/Tested (%)*	*p*-Value	OR (CI _95%_)	*N° Pos/Tested (%)*	*p*-Value	OR (CI _95%_)
A1	8/31 (25.8)	<0.001	2.26 (0.6–8.5)	13/31 (41.9)	<0.001	10.11 (2.04–50.19)	1/31 (3.2)	<0.001	1.00 (Reference group)
A2	4/30 (13.3)		1.00 (Reference group)	24/30 (80)		56.00 (10.33–303.68)	2/30 (6.7)		2.14 (0.18–24.96)
B1	16/30 (53.3)		7.43 (2.08–26.55)	2/30 (6.7)		1.00 (Reference group)	11/30 (36.7)		17.37 (2.07–145.61)
B2	14/30 (46.7)		5.69 (1.59–20.33)	12/30 (40)		9.33 (1.87–46.68)	0/30 (0)		NA
C1	9/26 (34.6)		3.44 (0.91–12.97)	14/26 (54.8)		16.33 (3.2–83.25)	0/26 (0)		NA
C2	26/30 (86.7)		42.25 (9.53–187.22)	4/30 (13.3)		2.15 (0.36–12.76)	0/30 (0)		NA
Total	77/177 (43.5)			69/177 (38.9)			14/177 (7.9)		

Dependent variable is *Campylobacter* spp. positive/negative status. OR: odds ratio, CI_95%_: 95% confidence interval, NA: not applicable due to zero positive samples.

**Table 4 animals-12-02978-t004:** Antibiotic resistance detected in *C. jejuni* and *C. coli* strains isolated from laying hens housed in enriched cages (A), barns (B) and aviaries (C).

			AZM		CHL		CIP		ENR		E		CN		NA		TE		SXT	
	Flock (N° strains)		I	R	I	R	I	R	I	R	I	R	I	R	I	R	I	R	I	R
*C. jejuni*	Cage (A)	A1 (8)	0 (0)	0 (0)	0 (0)	0 (0)	1 (12.5)	5 (62.5)	0 (0)	5 (62.5)	0 (0)	0 (0)	0 (0)	0 (0)	0 (0)	3 (37.5)	0 (0)	0 (0)	0 (0)	0 (0)
		A2 (4)	0 (0)	0 (0)	0 (0)	0 (0)	0 (0)	0 (0)	0 (0)	0 (0)	0 (0)	0 (0)	0 (0)	0 (0)	0 (0)	0 (0)	0 (0)	0 (0)	0 (0)	0 (0)
		Sub total (12)	0 (0)	0 (0)	0 (0)	0 (0)	1 (8.3)	5 (41.6)	0 (0)	5 (41.6)	0 (0)	0 (0)	0 (0)	0 (0)	0 (0)	3 (25)	0 (0)	0 (0)	0 (0)	0 (0)
	Barn (B)	B1 (15)	0 (0)	0 (0)	0 (0)	0 (0)	0 (0)	9 (60)	1 (6.7)	4 (26.7)	0 (0)	0 (0)	0 (0)	0 (0)	0 (0)	6 (40)	0 (0)	4 (26.7)	0 (0)	8 (53.3)
		B2 (14)	0 (0)	0 (0)	0 (0)	1 (7.1)	2 (14.3)	0 (0)	0 (0)	0 (0)	0 (0)	1 (7.1)	0 (0)	0 (0)	0 (0)	1 (7.1)	0 (0)	0 (0)	0 (0)	2 (14.3)
		Sub total (29)	0 (0)	0 (0)	0 (0)	0 (0)	2 (6.9)	9 (31)	1 (3.4)	4 (13.8)	0 (0)	1 (3.4)	0 (0)	0 (0)	0 (0)	7 (24.1)	0 (0)	4 (13.8)	0 (0)	10 (34.5)
	Aviary (C)	C1 (9)	0 (0)	0 (0)	0 (0)	0 (0)	0 (0)	0 (0)	0 (0)	0 (0)	0 (0)	0 (0)	0 (0)	0 (0)	0 (0)	0 (0)	0 (0)	0 (0)	0 (0)	0 (0)
		C2 (26)	0 (0)	0 (0)	0 (0)	0 (0)	0 (0)	0 (0)	0 (0)	0 (0)	0 (0)	0 (0)	0 (0)	0 (0)	0 (0)	0 (0)	0 (0)	0 (0)	0 (0)	0 (0)
		Sub total (35)	0 (0)	0 (0)	0 (0)	0 (0)	0 (0)	0 (0)	0 (0)	0 (0)	0 (0)	0 (0)	0 (0)	0 (0)	0 (0)	0 (0)	0 (0)	0 (0)	0 (0)	0 (0)
	Total (76)		0 (0)	0 (0)	0 (0)	0 (0)	3 (3.9)	14 (18.4)	1 (1.3)	9 (11.8)	0 (0)	1 (1.3)	0 (0)	0 (0)	0 (0)	10 (13.1)	0 (0)	4 (5.3)	0 (0)	10 (13.1)
*C. coli*	Cage (A)	A1 (13)	0 (0)	0 (0)	0 (0)	0 (0)	4 (30.8)	2 (15.4)	0 (0)	2 (15.4)	0 (0)	0 (0)	0 (0)	0 (0)	2 (15.4)	2 (15.4)	0 (0)	2 (15.4)	2 (15.4)	0 (0)
		A2 (20)	0 (0)	1 (5)	0 (0)	0 (0)	4 (20)	1 (5)	0 (0)	1 (5)	0 (0)	0 (0)	0 (0)	0 (0)	1 (5)	1 (5)	0 (0)	2 (10)	1 (5)	5 (25)
		Sub total (33)	0 (0)	1 (3)	0 (0)	0 (0)	8 (24.2)	3 (9)	0 (0)	3 (9)	0 (0)	0 (0)	0 (0)	0 (0)	3 (9)	3 (9)	0 (0)	4 (12.1)	3 (9)	5 (15.1)
	Barn (B)	B1 (2)	0 (0)	0 (0)	0 (0)	0 (0)	0 (0)	2 (100)	0 (0)	1 (50)	0 (0)	0 (0)	0 (0)	0 (0)	1 (50)	1 (50)	0 (0)	2 (100)	1 (50)	1 (50)
		B2 (12)	0 (0)	0 (0)	0 (0)	0 (0)	0 (0)	11 (91.7)	3 (25)	2 (16.7)	0 (0)	0 (0)	0 (0)	0 (0)	4 (33.3)	3 (25)	0 (0)	7 (58.3)	4 (33.3)	0 (0)
		Sub total (14)	0 (0)	0 (0)	0 (0)	0 (0)	0 (0)	13 (92.8)	3 (21.4)	3 (21.4)	0 (0)	0 (0)	0 (0)	0 (0)	5 (35.7)	4 (28.6)	0 (0)	9 (64.3)	5 (35.7)	1 (7.1)
	Aviary (C)	C1 (12)	0 (0)	0 (0)	0 (0)	0 (0)	0 (0)	9 (75)	2 (16.7)	4 (33.3)	0 (0)	0 (0)	0 (0)	0 (0)	2 (16.7)	4 (33.3)	0 (0)	9 (75)	4 (33.3)	0 (0)
		C2 (4)	0 (0)	0 (0)	0 (0)	0 (0)	0 (0)	1 (25)	1 (25)	0 (0)	0 (0)	0 (0)	0 (0)	0 (0)	1 (25)	0 (0)	0 (0)	3 (75)	1 (25)	0 (0)
		Sub total (16)	0 (0)	0 (0)	0 (0)	0 (0)	0 (0)	10 (62.5)	3 (18.7)	4 (25)	0 (0)	0 (0)	0 (0)	0 (0)	3 (18.7)	4 (25)	0 (0)	12 (75)	5 (31.2)	0 (0)
	Total (63)		0 (0)	1 (1.6)	0 (0)	0 (0)	8 (12.7)	13 (20.6)	16 (25.4)	10 (15.9)	0 (0)	0 (0)	0 (0)	0 (0)	11 (17.5)	11 (17.5)	0 (0)	25 (39.7)	13 (20.6)	6 (9.5)

The cutoff values as defined by EUCAST (European Committee on Antimicrobial Susceptibility Testing) were used to interpret the antimicrobial susceptibilities to AZM: azithromycin; CIP: ciprofloxacin; E: erythromycin; TE: tetracycline. The cutoff values as defined by CLSI Clinical Laboratory and Standards Institute) were used the antimicrobial susceptibilities to CN: gentamicin; NA: nalidixic acid; SXT: trimethoprim-sulfamethoxazole CHL: chloramphenicol; ENR: enrofloxacin.

**Table 5 animals-12-02978-t005:** Multidrug resistance found in *C. jejuni* and *C. coli* strains isolated from laying hens housed in enriched cages (A), barns (B) and aviaries (C).

			3 Drugs			4 Drugs			
		Flock	CIP/ENR/NA	CIP/NA/ SXT	CIP/ENR/ SXT	CIP/ENR/ NA/TE	CIP/ENR/NA/SXT	CIP/NA/ TE/SXT	CHL/E/NA/SXT
Multidrug resistant *C. jejuni:* 15/76 (19.7%)	Cage (A)	A1	3/8 (37,5)	0/8 (0)	0/8 (0)	0/8 (0)	0/8 (0)	0/8 (0)	0/8 (0)
		A2	0/4 (0)	1/4 (25)	0/4 (0)	0/4 (0)	1/4 (25)	0/4 (0)	0/4 (0)
		Sub total	3/12 (25)	1/12 (8.3)	0/12 (0)	0/12 (0)	1/12 (8.3)	0/12 (0)	0/12 (0)
	On floor (B)	B1	0/15 (0)	2/15 (13.3)	2/15 (13.3)	0/15 (0)	2/15 (13.3)	2/15 (13.3)	0/15 (0)
		B2	0/14 (0)	0/14 (0)	0/14 (0)	0/14 (0)	0/14 (0)	0/14 (0)	1/15 (7.1)
		Sub total	0/29 (0)	2/29 (6.9)	2/29 (6.9)	0/29 (0)	2/29 (6.9)	2/29 (6.9)	1/29 (3.4)
	Aviary (C)	C1	0/9 (0)	0/9 (0)	0/9 (0)	0/9 (0)	0/9 (0)	0/9 (0)	0/9 (0)
		C2	0/26 (0)	0/26 (0)	0/26 (0)	0/26 (0)	0/26 (0)	0/26 (0)	0/26 (0)
		Sub total	0/35 (0)	0/35 (0)	0/35 (0)	0/35 (0)	0/35 (0)	0/35 (0)	0/35 (0)
		Total	3/76 (3.9)	3/76 (3.9)	2/76 (2.6)	0/76 (0)	3/76 (3.9)	3/76 (3.9)	1/76 (1.3)
Multidrug resistant *C. coli: 11/63 (17.5%)*	Cage (A)	Flock A1	2/13 (15.4)	0/13 (0)	0/13 (0)	0/13 (0)	0/13 (0)	0/13 (0)	0/13 (0)
		Flock A2	3/20 (15)	0/20 (0)	0/20 (0)	0/20 (0)	0/20 (0)	0/20 (0)	0/20 (0)
		Sub total	5/33 (15.1)	0/33 (0)	0/33 (0)	0/33 (0)	0/33 (0)	0/33 (0)	0/33 (0)
	On floor (B)	B1	0/2 (0)	0/2 (0)	0/2 (0)	1/2 (50)	0/2 (0)	0/2 (0)	0/2 (0)
		B2	0/12 (0)	0/12 (0)	0/12 (0)	1/12 (8.3)	0/12 (0)	0/12 (0)	0/12 (0)
		Sub total	0/14 (0)	0/14 (0)	0/14 (0)	2/14 (14.3)	0/14 (0)	0/14 (0)	0/14 (0)
	Aviary (C)	C1	2/12 (16.7)	0/12 (0)	0/12 (0)	2/12 (16.7)	0/12 (0)	0/12 (0)	0/12 (0)
		C2	0/4 (0)	0/4 (0)	0/4 (0)	0/4 (0)	0/4 (0)	0/4 (0)	0/4 (0)
		Sub total	2/16 (12.5)	0/16 (0)	0/16 (0)	2/16 (12.5)	0/16 (0)	0/16 (0)	0/16 (0)
		Total	7/63 (11.1)	0/63 (0)	0/63 (0)	4/63 (6.3)	0/63 (0)	0/63 (0)	0/63 (0)

The cutoff values as defined by EUCAST (European Committee on Antimicrobial Susceptibility Testing) were used to interpret the antimicrobial susceptibilities to AZM: azithromycin; CIP: ciprofloxacin; E: erythromycin; TE: tetracycline. The cutoff values as defined by CLSI Clinical Laboratory and Standards Institute) were used the antimicrobial susceptibilities to CN: gentamicin; NA: nalidixic acid; SXT: trimethoprim-sulfamethoxazole CHL: chloramphenicol; ENR: enrofloxacin.

## Data Availability

Data is contained within the article.

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
