# Peer review of "Prevalence and Antimicrobial Resistance of *Campylobacter jejuni* and *Campylobacter coli* from Laying Hens Housed in Different Rearing Systems"

_animals, 2022, doi:10.3390/ani12212978_

Round 1

Reviewer 1 Report

The study by Casalino et al. investigated the incidence of C. jejuni and C. coli infection and determined the antimicrobial resistance of the detected Campylobacter spp in laying hen flocks housed in different rearing systems. In addition, the corticosterone and IL- 6 levels were measured.

The study found that 77/177 (43.5%) and 69/177 (38.95) laying hens harboured C. jejuni and C. coli, respectively.  Antimicrobial resistance was reported toward ciprofloxacin, enrofloxacin, and nalidixic acid in both strains, with high frequency observed in C. coli.

 The measurement of IL-6 and corticosterone showed the highest level in laying hens housed in aviaries compared to the laying hens housed in the barn and the cage systems.

This study is very interesting, relevant, and well-written. But some minor issues must be corrected before considering this manuscript for publication.

Specific comments:

Line 20: detected in 14 animals? Laying hens or birds? Please use the consistent word throughout the manuscript.

Line 46: Please add specific details to the sentence “Multidrug resistance was also found”

Line 186: OXOID, please use lowercase, also add the city of manufacturing

Line 189: Remove bold from (Campylobacter agar base).

Line 315: in table 5. Multi-drug resistance C. coli, please add the details number of the resistance.   Resistance to 2 drugs considered MDR? based on what decided the MDR?

Line 415: please remove the dash-line after the word ''barn''

What is the correlation between the high level of IL-6 and corticosterone and the incidence of Campylobacter spp in laying hens?

 Did the author use positive control when performing antimicrobial testing?

Author Response

Dear reviewer 1,

thank you for your comments and suggestions. The manuscript has been modified, accordingly. Below, I reported the replies to your comments. The changes according to your suggestions are coloured in yellow in the manuscript.

Best regards

The study by Casalino et al. investigated the incidence of C. jejuni and C. coli infection and determined the antimicrobial resistance of the detected Campylobacter spp in laying hen flocks housed in different rearing systems. In addition, the corticosterone and IL- 6 levels were measured.

The study found that 77/177 (43.5%) and 69/177 (38.95) laying hens harboured C. jejuni and C. coli, respectively.  Antimicrobial resistance was reported toward ciprofloxacin, enrofloxacin, and nalidixic acid in both strains, with high frequency observed in C. coli.

 The measurement of IL-6 and corticosterone showed the highest level in laying hens housed in aviaries compared to the laying hens housed in the barn and the cage systems.

This study is very interesting, relevant, and well-written. But some minor issues must be corrected before considering this manuscript for publication.

Comment. Line 20: detected in 14 animals? Laying hens or birds? Please use the consistent word throughout the manuscript.

Reply. The data are referred to laying hens. We corrected the used terms.

Comment.Line 46: Please add specific details to the sentence “Multidrug resistance was also found”

Reply: Details have been provided according to your suggestion.

Comment. Line 186: OXOID, please use lowercase, also add the city of manufacturing

Reply: It has been corrected.

Comment.Line 189: Remove bold from (Campylobacter agar base).

Reply: It has been removed.

Comment.Line 315: in table 5. Multi-drug resistance C. coli, please add the details number of the resistance

Reply. Detail number of resistance about C. coli has been added in table 5.

Comment. Resistance to 2 drugs considered MDR? based on what decided the MDR?

Reply. Thank you for this suggestion. it is more correct to consider the resistance to more than two molecules as MDR. We modified table 5 and corrected the rates of MDR in the text of manuscript, accordingly.

Comment. Line 415: please remove the dash-line after the word ''barn''

Reply: It has been corrected.

Comment.What is the correlation between the high level of IL-6 and corticosterone and the incidence of Campylobacter spp in laying hens?

Reply. Based on the results of Campylobacter detection and levels of plasma stress indicators values found in the differently reared flocks, a correlation seems to be present, probably due to a major spread of bacteria by infected animals in the flock (with a consequent higher prevalence of infection) and stressful conditions. This is a hypothesis, but it needs further investigations as reported in the text of manuscript (Lines 385-395).

Comment.  Did the author use positive control when performing antimicrobial testing?

Reply. Thank you for this suggestion. We forgot to mention of the two positive controls used for the identification of Campylobacter strains. They were C. jejuni ATCC 29428 and C. coli ATCC 33559 (LGC Promochem,Teddington, UK). We added the information in the text of the manuscript (Lines 214-216) and a figure (Figure 1) showing the identification of C. jejuni and C. coli strains by multiplex PCR (Lines 217-224).

Reviewer 2 Report

Casalino et al. studied the prevalence and antimicrobial resistance of Campylobacter jejuni and Campylobacter coli from laying hens housed in different rearing systems. Authors found resistance to quinolones, tetracycline and trimethoprim/sulfamethoxazole, as well as multidrug resistance in the tested strains. The study is very interesting and deserve to be published, however, a revision is required.

1.      Bacterial species should be written italic throughout the manuscript, even in the „References“ section. please see the attached file

2.      Tables 4, and 5: I propose to use these tables as Appendix, and present them here collectively/as simple tables. Also please describe the tables to stand alone: All abbreviations should be described in the table legend.

3.      Abbreviations should be revised throughout the manuscript, please use the full names for the first time, then use the abbreviation. Please see the attached file.

4.      Lines 155-163: I propose to link this paragraph with campylobacter and transfer it before the aim of the work.

5.      Material and methods: Please provide more data about all chemicals, media, and kits used in this study: "Supplier, City, and Country", please see the attached file. 

Author Response

Dear reviewer 2,

thank you for your comments. The manuscript has been accordingly modified. Below, I reported the replies to your comments. The changes according to your suggestion are coloured in light blue and tracked in the manuscript by “Track changes” function of word system.

Best regards

Casalino et al. studied the prevalence and antimicrobial resistance of Campylobacter jejuni and Campylobacter coli from laying hens housed in different rearing systems. Authors found resistance to quinolones, tetracycline and trimethoprim/sulfamethoxazole, as well as multidrug resistance in the tested strains. The study is very interesting and deserve to be published, however, a revision is required.

Comment. Bacterial species should be written italic throughout the manuscript, even in the „References“ section. please see the attached file

Reply. We provided to correct them.

Comment. Tables 4, and 5: I propose to use these tables as Appendix, and present them here collectively/as simple tables.

Reply. Please, let us leave the tables as whole, because the results have been analysed in detail about the various flocks based on rearing system. Therefore, the manuscript is clearer if it is possible reading all data in tables immediately, in our opinion.

Comment. Also please describe the tables to stand alone: All abbreviations should be described in the table legend.

Reply. We added a legend of the used abbreviations at the bottom of tables 4 and 5.

Reviewer 3 Report

Dear authors

You provided an original manuscript including a relevant field study for the isolation and phenotypic characterization of Camýlobacter coli and Campylobacter jejuni in three different layer housing with a well-performed statistical analysis. Minor editing is suggested, as follows:

1) L81 Please change from "Salmonella Enterica sub. Enterica" to "Salmonella enterica subsp. enterica"

2) L81 and L82 Please use "Escherichia coli" instead of "Escherichia (E.) coli"

3) L87 Please add "the" before "Campylobacter". 

4) L145 and L146 "... In positive farms, Campylobacter or DNA Campylobacter is detected in..." "In positive farms, the isolation of Campylobacter spp. or the detection of DNA of Campylobacter can be made in..."

5) L154 to L163 This paragraph needs to be moved before the sentence of the aims of this work.

6) L166 Please change from "... laying hen´s farms..." to "... laying hen farms...".

7) L167 Please delete "farm" changing from "(farm A)..." to "(A)".

8) L777 Please change from "Animal Based Measures" to "animal based measures".

9) L178 Please change from "Interleukin" to "interleukin"

10) L182 to L183 How much time did you use to transport the refrigerated samples from the field to the Department of Veterinary Medicine of Bari.

11) L186 The information of the manufacturer needs to be completed. Please include the city between the name of the manufacturer (Oxoid) and country (UK).

12) L187, L189-L191, and L194 The term "OXOID" needs to be changed to "Oxoid".

13) L189 Please add "Then," at the beginning of the sentence and change from "... Campylobacter agar base... " to "...Campylobacter agar base...". 

14) L192 Please include "above described" before "same".

15) L193 Please add "plates" after "blood agar" and incubated at... for...

16) L195 Please add "of each suspected isolate" "DNA extraction".

17) L202 Please add "and" after "DNA," and change from "...for a final volume of 25 ul." to " to complete a total volume of 25 ul.".

18) L212 to L215 Please include the manufacturer/s, city/ies, and country/ies of each of the antibiotic disks included in this paragraph.

19) L219 The subheading must include "ELISA" instead of "Elisa".

20) L223 Please delete ", San Diego, CA, USA)".

21) L233 Please delete "... of bird..."

22) L296 and L315 Please complete the headings of Tables 4 and 5 following a similar style to the heading of Table 3.

23) L298 to L304 and L317 to L319 Please References related to the names of the antibiotics, reference values of antibiotic resistance, and which standardized methodologies were followed (e.g. European Committee for Antimicrobial Susceptibility Testing)

24) L308 to L314 Please include all full names of the studied antibiotics as previously written in the manuscript to avoid misunderstandings.

25) L442 to L444 Please delete from ", such as" to "baths," because it is redundant. 

Author Response

Dear reviewer 3,

thank you for your comments. The manuscript has been accordingly modified. Below, I reported the replies to your comments. The changes according to your suggestion are coloured in green and tracked in the manuscript by “Track changes” function of word system.

Best regards

Gaia Casalino

Reviewer 3

You provided an original manuscript including a relevant field study for the isolation and phenotypic characterization of Camýlobacter coli and Campylobacter jejuni in three different layer housing with a well-performed statistical analysis. Minor editing is suggested, as follows:

Comment. L81 Please change from "Salmonella Enterica sub. Enterica" to "Salmonella enterica subsp. enterica"

Reply. It has been corrected in the text of the manuscript

Comment. L81 and L82 Please use "Escherichia coli" instead of "Escherichia (E.) coli"

Reply. It has been corrected according to your suggestion.

Comment. L87 Please add "the" before "Campylobacter". 

Reply. It has been added.

Comment.L145 and L146 "... In positive farms, Campylobacter or DNA Campylobacter is detected in..." "In positive farms, the isolation of Campylobacter spp. or the detection of DNA of Campylobacter can be made in..."

Reply. We provided to change the sentence as you suggested.

Comment. L154 to L163 This paragraph needs to be moved before the sentence of the aims of this work.

Reply. According to your suggestion, we provided to move the paragraph before the aims of the study.

Comment. L166 Please change from "... laying hen´s farms..." to "... laying hen farms...".

Reply. It has been corrected.

Comment. L167 Please delete "farm" changing from "(farm A)..." to "(A)".

Reply. It has been corrected.

Comment. L777 Please change from "Animal Based Measures" to "animal based measures".

Reply. It has been corrected.

Comment. L178 Please change from "Interleukin" to "interleukin"

Reply. It has been corrected.

Comment. L182 to L183 How much time did you use to transport the refrigerated samples from the field to the Department of Veterinary Medicine of Bari.

Reply. The samples were transferred about within two hours after sampling, just the time that was necessary to travel from the farm to the Department. We specified the information in the text of manuscript (Line 183)

Comment. L186 The information of the manufacturer needs to be completed. Please include the city between the name of the manufacturer (Oxoid) and country (UK).

Reply. The required information has been added

Comment. L187, L189-L191, and L194 The term "OXOID" needs to be changed to "Oxoid".

Reply. All changes have been provided.

Comment. L189 Please add "Then," at the beginning of the sentence and change from "... Campylobacter agar base... " to "...Campylobacter agar base...". 

Reply. The changes have been provided in the manuscript

Comment. L192 Please include "above described" before "same".

Reply. We have included it (Lines 195-196).

Comment. L193 Please add "plates" after "blood agar" and incubated at... for...

Reply. The sentence has been corrected according to your suggestion

Comment. L195 Please add "of each suspected isolate" "DNA extraction".

Reply. It has been added.

Comment. L202 Please add "and" after "DNA," and change from "...for a final volume of 25 ul." to " to complete a total volume of 25 ul.".

Reply. We changed the sentence according to your suggestion.

Comment. L212 to L215 Please include the manufacturer/s, city/ies, and country/ies of each of the antibiotic disks included in this paragraph.

Reply. We added the required information.

Comment. L219 The subheading must include "ELISA" instead of "Elisa".

Reply. We made the correction.

Comment. L223 Please delete ", San Diego, CA, USA)".

Reply. We deleted it.

Comment. L233 Please delete "... of bird..."

Reply. We have deleted it.

Comment. L296 and L315 Please complete the headings of Tables 4 and 5 following a similar style to the heading of Table 3.

Reply. The headings of Tables 4 and 5 have been completed according to your suggestion.

Comment. L298 to L304 and L317 to L319 Please References related to the names of the antibiotics, reference values of antibiotic resistance, and which standardized methodologies were followed (e.g. European Committee for Antimicrobial Susceptibility Testing)

Thank you for the suggestion. References related to followed methodology and the used cut-off values to interpret the antimicrobial susceptibilities have been reported in the legend at the bottom of tables 4 and 5.

Comment. L308 to L314 Please include all full names of the studied antibiotics as previously written in the manuscript to avoid misunderstandings

Reply. We have included all full names of the studied antibiotics and we added a legend of the used abbreviations at the bottom of tables 4 and 5

Comment. L442 to L444 Please delete from ", such as" to "baths," because it is redundant.

Reply. We delete “such as” from the text. Regarding "baths," we retain the opportunity to leave the term in the text because it is referred to baths with disinfectant used in farms to disinfect the boots.

Round 2

Reviewer 2 Report

Many thanks again to the authors for this interesting manuscript and for the revision.  I agree to leave the tables as it is according to the authors' desire, although I have a concern about these big tables 

However, I still have minor comments from the first revision, that maybe not be noticed. I marked them again in the attached file in red color.  Please see the attached file

Author Response

Dear reviewer 2,

We are very sorry for the inconvenience. Now, we completed all changes according to your suggestion. They are coloured in orange in the revised version of the manuscript. We renumbered the references accordingly.

Best regards

Gaia Casalino
